# Effect of circle, surface type and stride duration on vertical head and pelvis movement in riding horses with pre-existing movement asymmetries in trot

**Eva Marunova**[1] *, **Elin Hernlund**[2], **Emma Persson-Sjödin**[2]

1 Department of Clinical Science and Services, The Royal Veterinary College, Hatfield, United Kingdom,
2 Department of Animal Biosciences, Swedish University of Agricultural Sciences, Uppsala, Sweden

* emarunova19@rvc.ac.uk

## Abstract

Head and pelvis vertical movement asymmetries in horses are often evaluated under different conditions yet better understanding is required of how these asymmetries are altered by factors such as surface type or circle size. This study investigated how stride duration, surface and lungeing in circles of different sizes influenced objectively measured head and pelvis movement asymmetries in riding horses in full training. Movement asymmetries were recorded with body mounted accelerometers and were based on the differences between the two vertical displacement minima or maxima of head (HDmin, HDmax) and pelvis (PDmin, PDmax) within a stride cycle. Each horse was evaluated during straight-line trot and during lungeing (d = 10m/15m) on hard and soft surfaces at slow and fast speed (determined by stride duration). All horses (N = 76) had at least one movement asymmetry parameter above a predefined thresholds (|HDmin| or |HDmax| >6mm, |PDmin| or |PDmax| >3mm) during a straight line trot on hard surface (baseline). The horses were assigned to a 'predominant asymmetry' group (HDmin, HDmax, PDmin, PDmax) based on which movement asymmetry parameter was the greatest during the baseline condition; the head movement asymmetry values were divided by two to account for the difference in magnitude of the thresholds. Analysis was carried out for each predominant asymmetry group separately using linear mixed models—outcome variable: predominant asymmetry parameter; random factor: horse; fixed factors: surface, direction with stride duration as covariate (P<0.05, Bonferroni post-hoc correction). The 'direction' conditions were either a straight-line locomotion ('straight') or lungeing with lungeing conditions further classified by circle diameter and by whether the limb which the predominant asymmetry was assigned to ('assigned limb') was on the inside or outside of the circle ('inside10', 'inside15', 'outside10', 'outside15'). Only parameters related to asymmetrical weight-bearing between contralateral limbs (HDmin, PDmin) were affected by changes in stride duration–the most common pattern was an increase in asymmetries as stride duration decreased. Only pelvic movement asymmetries were affected by lungeing. When the assigned hindlimb was on the inside of the circle, the PDmin asymmetries increased and PDmax asymmetries decreased compared to the straight-line condition. With the assigned hindlimb on the outside, PDmin asymmetries

**Data Availability Statement:** All relevant data are within the manuscript and its Supporting Information files.

**Funding:** The author(s) received no specific funding for this work.

**Competing interests:** Data used in this study was originally collected for a previous study, Effect of meloxicam treatment on movement asymmetry in riding horses in training. doi:10.1371/journal.pone. 0221117. Co-authors Elin Hernlund and Emma Persson-Sjodin co-authored this previous study. However, they were not the recipients of the grant that supported data collection for the previous study. Grants from The Swedish-Norwegian Foundation for Equine Research (grant number H1347029) and the Swedish Research Council Formas (grant number 2014-1200328225-26) supported collection of data for the previous study. This does not alter our adherence to PLOS ONE policies on sharing data and materials.

decreased but PDmax asymmetries did not change. Trotting on 10 m circle compared to 15 m circle did not increase movement asymmetries. In conclusion, circular motion and changes in stride duration altered movement asymmetries identified in horses in full ridden work but no changes were seen between the soft and hard surfaces. These patterns should be further investigated in clinically lame horses.

## Introduction

Clinical experience suggests that certain orthopaedic conditions can be exacerbated either on hard or soft ground or with the affected limb positioned on the inside or outside of a circle during lungeing [1]. However, even horses with symmetrical head and pelvis movement during a straight-line locomotion often display asymmetrical movement patterns during lungeing [2–5]. These asymmetries may be related to altered torso orientation on circles [6] and the differences in angulation and forces between the inside and outside limbs [7, 8]. Therefore, it is important to differentiate between lameness related asymmetries and normal head and pelvic adaptations as horses negotiate movement on a circle. Movement asymmetries are also affected by a circle size–as the circle diameter decreases, the body lean increases resulting in greater upper body movement asymmetries [9, 10]. Yet, the size of the circle is often not standardized so it important to understand whether small variation in the circle size would significantly influence movement asymmetries.

In horses with symmetrical movement during straight-line locomotion, the most consistent pattern observed during lungeing is an asymmetrical pelvis movement which resembles inside hindlimb weightbearing lameness [2, 4, 11]. This pattern was observed on both a soft surface [11] and a hard surface [2, 4, 11] suggesting that pelvic movement is more influenced by circle-related adaptations than a surface type. Head vertical displacement can also be influenced by circular motion [3, 11–13] but there is no clear pattern of circle-related head movement asymmetries arising from previous studies. This lack of agreement might be partially attributed to different surface types used during the evaluation as some studies used the same surface type for the straight-line assessment and lungeing [2, 11] while other studies used hard surface for straight-line evaluation and soft surface for lungeing [4, 12].

The surface type might be even more relevant in horses with movement asymmetries yet there is limited research on how head and pelvis movement asymmetries are affected by the type of surface. One previous study included horses with only head movement asymmetries but did not include any horses with only pelvic movement asymmetries [11]. Another study used horses with head and pelvic movement asymmetries [13] but the horses were not grouped for analysis by the most prominent type of asymmetry, i.e. forelimb or hindlimb asymmetries or whether the asymmetries were related to uneven weight-bearing or propulsion. Consequently, little is known about how weight-bearing and propulsion related movement asymmetries respond to the changes in the surface type.

Maintaining consistent trotting speed is recommended during multiple evaluations. However, speed can vary not only between repeated evaluations but also between straight-line and lungeing assessments [11, 13, 14] or between different types of surfaces [8]. This variation in speed can, in turn, influence movement asymmetries [15–18]. While speed is not easy to record in the clinical setting, stride duration can be readily obtained from sensor-based gait analysis systems and could be used as a proxy for speed due to inverse relationship between stride duration and speed [18–20].

Consequently, the aim of this study was to investigate how these surface and circle related factors influenced head and pelvis movement asymmetry parameters in a population of horses in full ridden work that presented with movement asymmetries but were considered sound by their owners. We hypothesised pelvic movement asymmetries will be affected by circular motion more than head movement asymmetries. In addition, we hypothesised that changes in movement asymmetries will be greater on a circle with decreasing circle diameter. It was also anticipated that movement asymmetries would increase with decreasing stride duration.

## Materials and methods

### Ethics and informed consent

The data used in the present study were collected as part of a study aiming to assess the effect of oral administration of non-steroidal anti-inflammatory drugs (NSAID) on upper body movement asymmetries [21] which was approved by the Ethical Committee for Animal Experiments, Uppsala, Sweden, application number C 48/13 and C 92/15. Informed written consent was obtained from all horse owners.

### Horses

A convenience sample of warmblood riding horses in Sweden was used; horses were either privately owned or they belonged to two equestrian centres and two riding schools. Horses were in full training, were considered free from lameness by their owners and had not been treated for lameness during the two months preceding data collection. Horses were included in the original study if they presented with at least one movement asymmetry parameter above thresholds recommended for the clinical use of commercially available sensor-based system (Lameness Locator, Equinosis, Columbia, MO, USA) during straight line trot in hand on the first day of the study (6mm for head movement asymmetries and 3mm for pelvis movement asymmetries).

If the horses were considered too lame to continue in their normal training (visual lameness grade greater than 2 degrees on a 0–5 ordinal scale [22]), they were excluded from the study subject to the judgment of the veterinarians conducting the original study [21]. However, a full lameness investigation was not performed as a part of the study.

**Instrumentation.**   The horses were instrumented with three inertial measurement units (IMU). One uni-axial gyroscope with a range of 300˚/s was attached with a specially designed pastern wrap to the dorsal side of the right forelimb pastern. Two uni-axial accelerometers with a range of 6 g were then attached, one to the poll with a felt head bumper and one taped to the midline between the two tubera sacrale. The sensors measured 3.2 x 3.0 x 2.0 cm and had a mass of 28 g. Data were digitally recorded (8 bits) at 200 Hz and wirelessly transmitted to a handheld computer.

### Data collection

In the original study, the horses were treated with NSAID or placebo for four days followed by a 14–16 day washout period before receiving the other treatment for four days. Objective movement asymmetry data were recorded at four points—on day one before each treatment and on day four of each treatment. For the present study, only data from days without active NSAID treatment were considered for inclusion (up to three data collection days per horse).

On each data collection day, movement asymmetry measurements were obtained under predefined exercise conditions. The horses were evaluated in trot in hand on a straight line and lunged in circles with a diameter of 10 m and 15 m on the left and right rein. Markers

were placed on the lunge line to indicate the size of the circle to ensure consistency between handlers and trials. The horses were also assessed on two different types of surface (hard and soft) except for 15 m circle which was only evaluated on a soft surface. The hard surface usually consisted of packed dirt or occasionally asphalt, and the soft surface was an arena surface (either sand, sand and woodchip or sand and fibre). Therefore, surface specifications could vary between horses but the surface was kept consistent for each individual horse throughout the study, i.e. they were trotting on the same soft and hard surface both in hand and during lungeing throughout the trial. Each horse was randomly allocated to start with either on a hard/soft surface, a left/right direction and circles/straight lines.

In addition, during each condition (for example lungeing to the left on a soft surface), the horses were evaluated at horse's preferred speed and then in faster trot (with the handler instructed to trot the horses faster). The stride duration obtained from the IMU system was then used to verify that two different speed categories were achieved (i.e. 'slow' was assigned to the measurement with higher stride duration).

## Data processing

**Movement asymmetry parameters.** The Lameness Locator software was used for analysis of the sensor data. Briefly, recorded vertical acceleration from the head and pelvis sensor was converted to vertical displacement as previously described by Keegan et al. [23] with stride splitting performed based on angular sagittal plane velocity data from the limb mounted sensor's gyroscope [24]. The mean of the stride-by-stride difference in the local displacement minima or maxima was calculated and the following four movement asymmetry parameters were tabulated for each exercise condition: the difference between local displacement minima for the head (HDmin) or pelvis (PDmin) during right and left limb stances, difference between local displacement maxima for the head (HDmax) or pelvis (PDmax) during suspension phase following right and left limb stance. The IMU system followed the usual sign convention with positive values indicating asymmetries attributed to the right limb and negative values representing asymmetries attributed to the left limb.

**Allocation to predominant movement asymmetry groups.** A predominant movement asymmetry was identified for each of the three data collection days–this was assigned based on the greatest movement asymmetry parameter during a straight line trot on a hard surface at slow speed which was chosen as a baseline exercise condition. To account for the difference in the magnitude of the threshold values for the head and pelvic movement asymmetry parameters, the head movement asymmetry values were divided by two to account for the difference in the magnitude of the threshold values for the head and pelvic movement asymmetry parameters: predominant asymmetry = Max $\left\{ \frac{|HDmin|}{2}, \frac{|HDmax|}{2}, |PDmin|, |PDmax| \right\}$. The baseline movement asymmetry had to be above the previously mentioned thresholds (6mm for head movement asymmetries, 3mm for pelvis movement asymmetries) and at least 1.25 time greater than the standard deviation to eliminate any horses with inconsistent movement asymmetries.

If a horse was showing the same type of predominant movement asymmetry across two or three data collection days (for example HDmin was identified as predominant movement asymmetry during pre-placebo and placebo days), only the data from the day with the highest movement asymmetry were included (Fig 1). If the horse was displaying different type of predominant movement asymmetry on different data collection days, the data from multiple days were included but in different groups. For example, a predominant HDmin asymmetry on a pre-placebo day and a predominant PDmin asymmetry on a placebo day, would result in inclusion of this horse in HDmin group (data from pre-placebo day) and in PDmin group (data from placebo day). Therefore, no horse was included more than once in the same

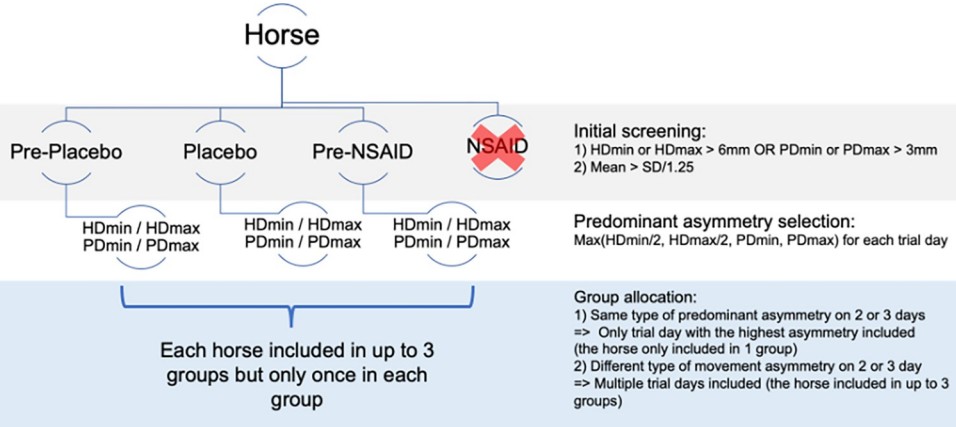

**Fig 1. Schematic representation of inclusion criteria and allocation to predominant asymmetry groups.** Grey shading–initial screening for consistent movement asymmetries above a predefined threshold on each data collection day. White shading–identification of a predominant asymmetry for each data collection day (one group of horses had placebo treatment first and the other group of horses underwent NSAID treatment first hence the order could vary between horses). Blue shading–decision process for allocation to movement asymmetry groups. HDmin or PDmin mean difference across run/trial between the two halves of a stride for vertical displacement minima of head (H) or pelvis (P), HDmax or PDmax mean difference across run/trial between the two halves of a stride for vertical displacement maxima of head (H) or pelvis (P).

predominant asymmetry group, but the same horse could be included in multiple predominant movement asymmetry groups with data from different days.

**Data normalisation.** The lungeing direction was relabelled with respect to the position of the limb to which the predominant asymmetry was assigned ('assigned limb'), i.e. inside or outside of a circle. This meant that for horses with predominant movement asymmetries associated with a right limb, lungeing to the right was relabelled as 'inside' and to the left as 'outside' and vice versa for horses with predominant movement asymmetries associated with a left limb. Furthermore, to combine data from horses with left and right limb related asymmetries into one model, the values of the predominant asymmetry parameters were normalised as if all the horses had right-limb predominant asymmetry. This meant that if the baseline value of the predominant asymmetry parameter was negative (for example, HDmin = -7 mm indicating a left forelimb weight-bearing deficit), the values of this parameter were multiplied by (-1) for all exercise conditions.

## Statistical analysis

All statistical analyses were performed in IBM SPSS (v28, IBM, Armonk, NY, USA). The level of significance was set at $p < 0.05$. Four separate linear mixed models were implemented with HDmin, HDmax, PDmin or PDmax as outcome variables, 'horse' as random factor, 'direction' (straight, inside10, inside15, outside10, outside15) and 'surface' (hard, soft) as fixed factors and 'stride duration' as covariate. All 2-way and 3-way interactions were initially included in all models. If the interactions were not significant, the model was reduced to include only the individual factors. Bonferroni corrections were implemented when investigating pairwise significant differences for the fixed factors. The histograms of the residuals were inspected for normality and the residual values were plotted against fitted values to assess heteroscedasticity. Normal distribution and homoskedasticity were confirmed for all mixed models used in this study.

**Table 1. Summary of the predominant movement asymmetries during straight line trot on a hard surface at horse's preferred speed (slow condition).**

| Group | Number of horses | Mean (mm) | Standard deviation | Range (mm) |
|---|---|---|---|---|
| HDmin | N = 25 | 11.3 | 5.1 | 6.1–29.9 |
| HDmax | N = 16 | 13.3 | 7.3 | 6.2–30.3 |
| PDmin | N = 39 | 6.4 | 3.7 | 3.3–23.1 |
| PDmax | N = 34 | 5.7 | 1.7 | 3.1–10.4 |

## Results

### Horses and baseline movement asymmetry

In total, 76 horses were included with 48 geldings and 28 mares, mean (range) age 12 years (3–22 years), height at withers 167cm (152–180 cm), body mass 600 kg (460–750 kg). Horses included consisted of privately owned horses (n = 31), horses owned by the National Equestrian Centre at Strömsholm (n = 19), by the Swedish Horse Guards Society (n = 14), and by two riding schools (n = 12). Horses were used for dressage (n = 23), show jumping (n = 23), eventing (n = 1) or were all-round horses (n = 26). All included horses were warmbloods (62 Swedish Warmbloods, 5 Dutch Warmbloods, 3 Hanoverian Warmbloods and 4 other warmbloods).

The allocation to groups based on the predominant movement asymmetry is summarised in Table 1. Thirty horses were included in two groups and four horses were included in three groups. The head movement was more variable between horses as indicated by larger standard deviations and range for HDmin and HDmax groups.

### Stride duration

Stride duration was obtained from IMU sensors as a proxy for speed. Overall, the mean stride duration was higher during lungeing (+14–47 ms compared to straight line) and on a soft surface (+7–17 ms compared to a hard surface), Table 2. Within the individual horses, the difference between the minimum and maximum stride duration recorded across all the exercise conditions was 68 ms to 227 ms (mean = 128 ms).

### Forelimb-related movement asymmetries

HDmin was only significantly influenced by stride duration (p = 0.01) and not by surface or direction, i.e. the HDmin asymmetries did not differ between a straight line trot and lungeing or between the two types of surface (Table 3). The estimate of the stride duration coefficient indicates that for lower stride duration the HDmin movement asymmetry increased across all conditions by 0.03 mm for each 1 ms decrease in stride duration (Table 3). This would result

**Table 2. Mean (standard deviation) stride duration in ms for each exercise condition for horses with predominant forelimb related movements asymmetries (HDmin and HDmax groups) and predominant hindlimb related movement asymmetries (PDmin and PDmax groups).**

| Group | | HDmin and HDmax groups | | PDmin and PDmax groups | |
|---|---|---|---|---|---|
| Surface | | Hard Surface | Soft Surface | Hard Surface | Soft Surface |
| Direction | Straight | 710.5 (46.7) | 717.0 (45.0) | 741.3 (54.2) | 757.3 (50.2) |
| | Inside10 | 753.1 (53.4) | 767.1 (51.2) | 756.9 (48.2) | 774.3 (46.7) |
| | Inside15 | N/A | 754.6 (46.7) | N/A | 761.8 (44.9) |
| | Outside10 | 750.8 (54.6) | 763.7 (52.3) | 758.8 (51.5) | 772.9 (47.6) |
| | Outside15 | N/A | 753.9 (49.8) | N/A | 759.0 (46.0) |

**Table 3. Summary of results for the linear mixed models for horses with predominant forelimb related movement asymmetries (HDmin and HDmax groups).**

| Group | Fixed factors | p-value | Exercise Condition | | Estimate (95% confidence interval) |
|---|---|---|---|---|---|
| HDmin | Intercept | **<0.001** | - | | 27.9 (13.2, 42.6) |
| (N = 25) | Direction | 0.12 | - | | |
| | Surface | 0.39 | - | | |
| | Stride duration | **0.01** | - | | -0.03 (-0.05, -0.005) |
| HDmax | Intercept | 0.17 | - | | |
| (N = 16) | Direction | 0.35 | - | | |
| | Surface | 0.31 | - | | |
| | Stride duration | 0.80 | - | | -0.008 (-0.08, 0.06) |
| | Direction * Surface[a] | **0.002** | Inside 10 | Hard | 11.4 (7.6, 15.1) |
| | | | | Soft | 6.5 (2.7, 10.3) |
| | | | Inside15 | Soft | 6.6 (2.8, 10.3) |
| | | | Outside10 | Hard | 7.4 (3.7, 11.2) |
| | | | | Soft | 10.4 (6.7, 14.1) |
| | | | Outside15 | Soft | 11.6 (7.8, 15.4) |
| | | | Straight | Soft | 11.3 (7.4, 15.3) |
| | | | Straight | Hard | 11.2 (7.2, 15.3) |
| | Direction *Surface *Stride duration | **0.03** | Inside10 | Hard | -0.03 (-0.1, 0.05) |
| | | | | Soft | 0.05 (-0.03, 0.1) |
| | | | Inside15 | Soft | 0.02 (-0.06, 0.1) |
| | | | Outside10 | Hard | 0.05 (-0.03, 0.1) |
| | | | | Soft | -0.09 (-0.09, 0.07) |
| | | | Outside15 | Soft | -0.01 (-0.01, 0.07) |
| | | | Straight | Hard | -0.03 (-0.1, 0.06) |
| | | | | Soft | 0 |

HDmin mean difference between the two halves of a stride for vertical displacement minima of head, HDmax mean difference between the two halves of a stride for vertical displacement maxima of head. Inside or outside refers to the position on the circle for the limb to which the predominant movement asymmetry was assigned to during straight line trot on hard surface at slow speed (higher stride duration), 10 or 15 refers to the diameter of the circle (in m). The estimates indicate movement asymmetry in mm except for terms which include stride duration where the values indicate mm change in movement asymmetry per ms change in stride duration.

[a] HDmax estimated marginal means evaluated at mean stride duration of 754ms

in up to 1.7mm increase in HDmin during lungeing due to an associated increase in stride duration.

Although the 3-way interaction of direction, surface and stride duration was significant in the HDmax model (p = 0.03), the confidence intervals of the coefficient estimates included zero (Table 3) and there was no apparent pattern of how surface type influenced HDmax asymmetries (Fig 2).

## Hindlimb-related movement asymmetries

PDmin asymmetries were influenced by stride duration but the pattern was not the same across all the direction conditions as indicated but the significant interaction between direction and stride duration (p = <0.001, Table 4). Per 1 ms decrease in stride duration, the PDmin asymmetries increased by 0.02 mm for a straight-line trot condition and by 0.04 mm when the assigned hindlimb was on the inside of a circle (Fig 3). When the assigned hindlimb was on the outside, the PDmin asymmetries decreased by 0.01 mm for each 1 ms decrease in stride duration. When evaluated at mean stride duration (754 ms), the PDmin asymmetries

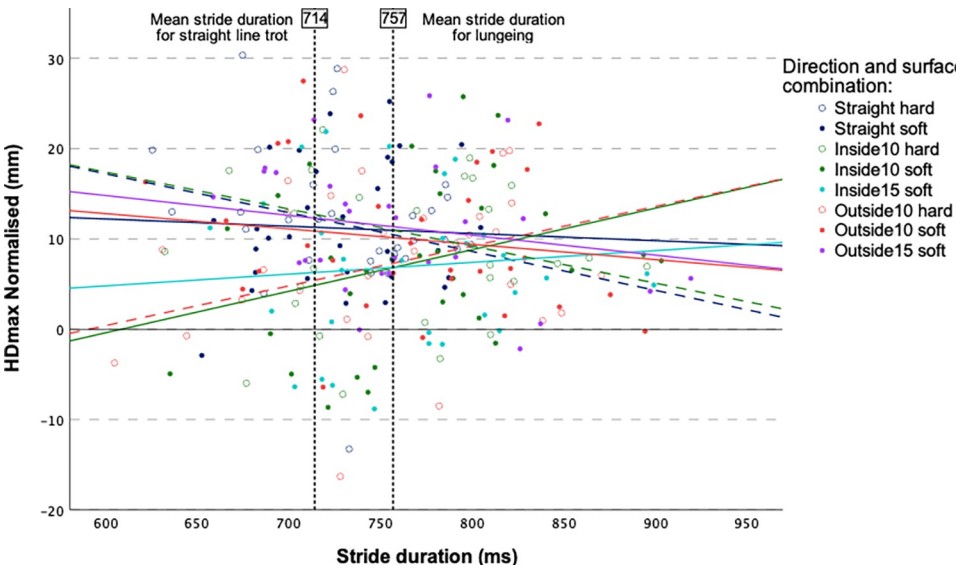

**Fig 2. Three-way interaction plot for direction and stride duration for horses with predominant HDmax movement asymmetries (N = 16 horses).** Solid lines represent soft surface condition and dashed lines represent hard surface condition. HDmax mean difference between the two halves of a stride for vertical displacement maxima of head. Inside or outside refers to the position on the circle for the limb to which the baseline movement asymmetry was assigned to during a straight-line trot on hard surface at slow speed. 10 or 15 refers to the diameter of the circle (in m).

were greater with the assigned hindlimb on the inside of a circle (10 mm) and smaller when the hindlimb was on the outside the circle (1.7–1.8 mm) in comparison to the straight-line trot (5.5 mm). There was no significant difference in PDmin asymmetries between lungeing on a 10 m or a 15 m circle and between the two types of surfaces.

PDmax asymmetries were significantly affected only by direction (p<0.001, Table 4). The PDmax asymmetries were lower when the assigned hindlimb was on the inside of a circle (2.9–3.4 mm) than when the assigned hindlimb was on the outside (5.4 mm). Compared to the straight-line trot, the PDmax asymmetries were lower (by 1.8 mm) when the assigned hindlimb was on the inside of a 10m circle.

## Discussion

This study investigated how stride duration, surface type and lungeing in circles of different sizes influenced objectively measured head and pelvis movement asymmetries in riding horses in full training. As expected, circular motion systematically affected pelvic movement asymmetries but not head movement asymmetries, i.e. the pelvic asymmetries differed between straight line trot and lungeing. On the other hand, the asymmetries were not different between the 10 m and 15 m circles or the two surface types. Consequently, movement asymmetries recorded in this population of horses in ridden work remained consistent despite some variation in circle size and surface type. However, the gait should be kept as consistent as possible between multiple evaluations as lower stride duration resulted in greater movement asymmetries.

### The effect of circular motion on movement asymmetries

Head and pelvic movement asymmetries during lungeing were previously observed even horses with symmetrical movement on a straight line lungeing [2–5]. These asymmetrical

**Table 4. Summary of results for the linear mixed models for horses with predominant hindlimb related movement asymmetries (PDmin and PDmax groups).**

| Group | Fixed factors | p-value | Exercise Condition | Estimate (95% confidence interval) |
|---|---|---|---|---|
| PDmin | Intercept | **<0.001** | - | 18.8 (8.8, 28.8) |
| (N = 39) | Direction[a] | **<0.001** | Inside10 | 10.0 (8.7, 11.2) |
| | | | Inside15 | 10.0 (8.6, 11.4) |
| | | | Outside10 | 1.8 (0.5, 3.0) |
| | | | Outside15 | 1.7 (0.3, 3.1) |
| | | | Straight | 5.5 (4.2, 6.8) |
| | Surface | 0.18 | - | |
| | Stride duration | **0.007** | - | -0.02 (-0.03, 0.004) |
| | Direction * Stride duration | **<0.001** | Inside10 | -0.02 (-0.04, -0.02) |
| | | | Inside15 | -0.02 (-0.04, -0.02) |
| | | | Outside10 | 0.03 (0.008, 0.04) |
| | | | Outside15 | 0.03 (0.006, 0.05) |
| | | | Straight | 0 |
| PDmax | Intercept | **0.001** | - | 10.5 (4.5, 16.5) |
| (N = 34) | Direction[a] | **<0.001** | Inside10 | 2.9 (1.9, 3.8) |
| | | | Inside15 | 3.4 (2.3, 4.6) |
| | | | Outside10 | 5.4 (4.4, 6.4) |
| | | | Outside15 | 5.4 (4.3, 6.6) |
| | | | Straight | 4.7 (3.7, 5.8) |
| | | | | Significantly different pairwise comparisons |
| | | | | (all p<0.001) |
| | | | | Straight-Inside10 |
| | | | | Inside10-Outside10 |
| | | | | Inside10-Outside15 |
| | | | | Inside15-Outside10 |
| | | | | Inside15-Outside15 |
| | Surface | 0.14 | - | |
| | Stride duration | 0.06 | - | |

PDmin mean difference between the two halves of a stride for vertical displacement minima of pelvis, PDmax mean difference between the two halves of a stride for vertical displacement maxima of pelvis. Inside or outside refers to the position on the circle for the limb to which the predominant movement asymmetry was assigned to during straight line trot on hard surface at slow speed (higher stride duration), 10 or 15 refers to the diameter of the circle (in m). The estimates indicate movement asymmetry in mm except for terms which include stride duration where the values indicate mm change in movement asymmetry per ms change in stride duration.
[a] PDmin and PDmax estimated marginal means evaluated at mean stride duration of 754 ms.

patterns could be attributed to the differences in angulation, forces and weight-bearing time between the inside and outside limbs [7, 8, 25]. In horses with movement asymmetries, these circle-related asymmetries can potentially increase or decrease pre-existing movement asymmetries depending on whether the assigned limb is on the inside or outside of a circle. Despite this, the head movement asymmetries did not increase during lungeing even though the horses in this study presented with asymmetries above thresholds recommended for clinical use. Same results were reported for reining horses in training, although that study only evaluated the horses on a soft surface [26]. In contrast, another study identified greater HDmin asymmetries when the assigned forelimb was on the inside of a circle during lunging on hard surface [11]. These contradicting results might be due to heterogenous population of horses with a varied degree of movement asymmetries. Although the horses used in this and the two earlier studies were in ridden work and were considered sound by their owners, a full lameness

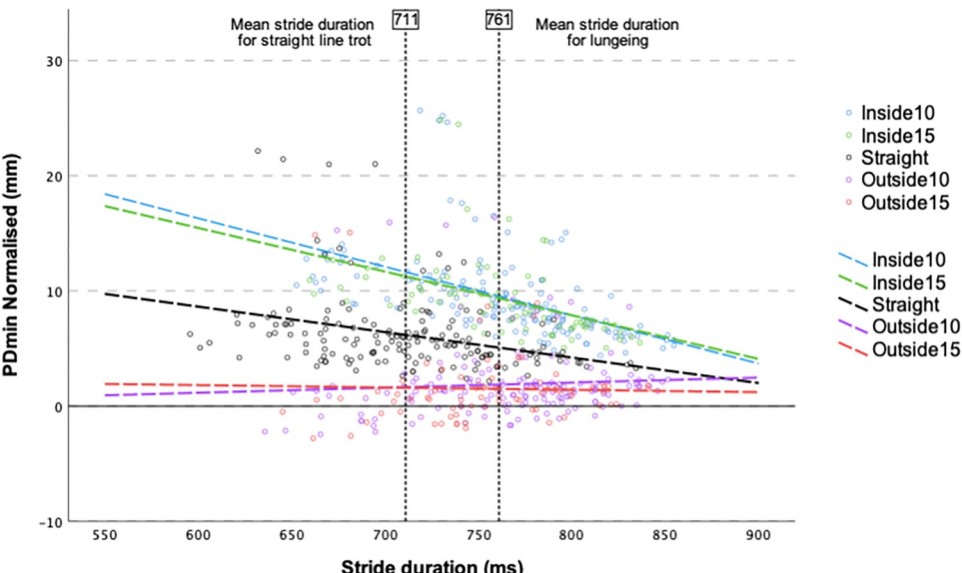

**Fig 3. Two-way interaction plot for direction and stride duration for horses with predominant PDmin movement asymmetries (N = 39 horses).** PDmin mean difference between the two halves of a stride for vertical displacement minima of pelvis. Inside or outside refers to the position on the circle for the hindlimb to which the baseline movement asymmetry was assigned to during a straight line trot on hard surface at slow speed. 10 or 15 refers to the diameter of the circle (in m).

examination was not carried out. Therefore, it is not possible to say whether any of these asymmetries were due to underlying pathologies. It is currently unknown whether asymmetries related to pain are affected by circular motion differently to asymmetries arising from laterality or normal biological variation. Consequently, further studies are warranted to establish how head movement asymmetries are affected by lungeing for specific forelimb lesions in clinically lame horses.

In agreement with our hypothesis, pelvis movement asymmetries were systematically influence by circular motion. The PDmin asymmetries increased when the assigned hindlimb was on the inside of a circle and decreased when the hindlimb was on the outside, a finding in line with previous studies [3, 12, 26, 27]. During lungeing, the pelvis is held more vertically (with less inward lean) during the inside hindlimb stance compared to the outside hindlimb stance [6]. This might explain why even horses with no pre-existing movement asymmetries display asymmetrical pelvic movement patterns during circular motion [2, 4, 26]. Consequently, the increase in PDmin asymmetries in one lungeing direction ('inside') and the decrease in the other direction ('outside') may be related to altered torso orientation on circles [6].

The PDmax asymmetries were also influenced by lungeing in this study but the only significant difference from a straight-line trot was a decrease in PDmax when the assigned hindlimb was on the inside of a 10-meter circle. On a circle, the outside hindlimb has to travel further [7] and the pelvis might rise less after the outside hindlimb stance in order to cover the increased horizontal distance [4]. In horses with PDmax asymmetries, this would results in a more symmetrical pelvis upward movement when the assigned hindlimb is positioned on the inside of a circle, as seen in this study and previously [26]. Nevertheless, the decrease in PDmax during lungeing in the present study was only small (1.8 mm) and lower than the repeatability thresholds previously established for this system [23] while the changes in PDmin asymmetries were above the repeatability threshold (>3 mm). Therefore, based on the results

from this and earlier studies [3, 12, 26, 27], it appears that circular motion affects asymmetries related to reduced weight-bearing more than those related to reduced propulsion in hindlimbs. Future studies should investigate how specific hindlimb orthopaedic conditions are affected by lungeing and the position of the limb on the circle.

Finally, as the circle size is often not standardized during lungeing evaluation, it is important to understand how variation in a circle size might influence movement asymmetries. In previous studies, a smaller circle diameter resulted in greater body lean and, consequently, greater upper body movement asymmetries [9, 10]. In the present study, however, no difference in movement asymmetries was seen between the two circle sizes. Consequently, the circles size might not have to be tightly controlled as small variation in a circle size (here a 5-meter difference in diameter) did not result in significant changes in movement asymmetries.

## The effect of surface type on movement asymmetries

Clinical experience suggests that lameness for certain orthopaedic conditions can be exacerbated on a soft or a hard surface [1]; for example, horses with foot pain often perform worse on a hard surface while horses with suspensory desmitis or tendonitis tend to be more lame on a soft surface. Yet, only small number of studies investigated how changing a surface type affects movement asymmetries [11, 13, 28].

Some influence of surface type was previously demonstrated in horses in ridden work with movement asymmetries above thresholds recommended for clinical use [11, 13]. Greater asymmetries were identified for HDmin on a soft surface during straight line trot and for PDmax on a hard surface during lungeing with the assigned hindlimb on the inside of a circle [13]. Although statistically significant, the size of the effect was small with only a 2 mm difference in asymmetries between the two surfaces. Given that these values are below the repeatability thresholds of IMU-based systems [23], such small changes might be simply due to normal variation between runs and not due to changes in the surface type. Indeed, this seems to be supported by the results from the present study as there was no difference in head and pelvic movement asymmetries between the evaluations on the soft and hard surfaces. In contrast, a much greater impact of a surface type was observed in another study with the HDmin asymmetries reaching much great values during lungeing on hard surface when the assigned forelimb was on the inside of a circle compared to the outside lungeing condition and lungeing on soft surface [11]. It should be noted that full clinical examination was not carried out in this or the earlier studies so it cannot be ruled out that some of the horses had underlying pathologies. This could have, in turn, influenced how the movement asymmetries changed on the different surfaces possibly explaining these contradicting results.

In clinically lame horses, only PDmax asymmetries were affected by surface type with greater changes identified on hard surface following successful diagnostic analgesia [28]. However, as diagnosis for these horses was not provided, it is not possible to say whether certain orthopaedic conditions result in greater asymmetries either on hard or soft surface. Further studies of clinically lame horses with confirmed diagnosis are warranted to better understand how the type of surface, possibly in combination with circular motion, influences movement asymmetries for different types of orthopaedic lesions.

## Considerations for repeated evaluations

Results from previous research suggest that movement asymmetries can increase with increasing speed [15, 17, 18]. This raises a question of how tightly the trotting speed needs to be controlled if multiple assessments are carried out and by how much the movement asymmetries

can vary due to inconsistencies in trotting speed. This is particularly relevant in the clinical practice as speed is rarely strictly controlled or recorded when horses are evaluated during multiple runs or under different conditions. In addition, surface type also influences speed as horses were shown to trot slower on soft artificial arena surface compared to hard surface [29] and in a sand arena compared to asphalt [30]. This is particularly relevant when comparing straight line and lungeing conditions as horses are often trotted in hand on a hard surface but lunged on a soft surface which makes comparison of movement asymmetries between the two conditions challenging.

In the present study, lower stride duration (a proxy for higher speed) resulted in greater HDmin asymmetries. This increase in HDmin asymmetries might be linked to increased load placed on the affected forelimb due to greater peak vertical forces in forelimbs at higher trotting speeds [19, 31], However, results from previous studies are somewhat contradicting. In forelimb lame horses evaluated on a treadmill, one study identified a positive association between speed and head movement asymmetries but only in horses with more severe forelimb lameness [15]. In contrast, another study found no effect of speed on head movement asymmetries in horses with mild to moderate forelimb lameness [18]. However, in the latter study, most horses presented with both forelimb and hindlimb lameness which might have influenced the results. For overground locomotion, positive relationship between head movement asymmetries and speed was only identified during lungeing but not during a straight-line assessment [17] which is only in partial agreement with the results from this study. In the present study, stride duration was used as a proxy for speed as stride duration can be readily obtained from IMU sensors. In general, as speed increases, the stride duration decreases [18–20]. However, stride duration can also change while the horse maintains the same speed, for example if they switch from high stride frequency and shorter strides to lower stride frequency and longer strides. Consequently, the results might vary if stride duration is used instead of direct speed measurement.

Changes in stride duration also influenced PDmin asymmetries in the present study. During a straight line trot, lower stride duration (a proxy for higher speed) resulted in greater PDmin asymmetries. Similarly, PDmin asymmetries were shown to increase with speed in hindlimb lame horses evaluated on a treadmill [18]. In contrast, PDmin asymmetries in subtly lame horses were not affected by speed during a straight-line trot evaluation [17]. On the other hand, the PDmin asymmetries increased with speed during lungeing which would suggest that keeping speed consistent might be more important for lungeing assessment. This is partially supported by the findings from the present study as the coefficient estimate for stride duration was greater for lungeing with the assigned limb on the inside of a circle compared to a straight-line trot. However, for the outside lungeing condition, the coefficient estimate was the smallest. Consequently, any given change in stride duration would result in proportionately greater changes in PDmin asymmetries during lungeing with the assigned limb on the inside compared to a straight-line trot and lungeing with the limb on the outside. These differences might be attributed to greater angulation and increased weight-bearing time of the inside limb during lungeing [7, 8, 25]. However, further studies are needed to better understand the mechanism of how horses adapt to changes in speed on a circle and how this impacts movement of the different upper body segments. Interestingly, HDmax and PDmax asymmetries in this study were not affected by changes in stride duration, a finding which is in line with previous research [17, 18, 26]. This would suggest that asymmetries related to propulsion deficits might not be affected by changes in speed or stride duration. This aspect warrants further investigation, especially in clinically lame horses with orthopaedic lesions that might with affect propulsion.

## Limitations

The aim of this study was to assess how these asymmetries, which are common in many horses in training, changed under different conditions on a given day. The horses included in this study were considered fit to continue in their regular ridden training by the study's veterinarian even though they were all presenting with a degree of movement asymmetries above thresholds that are often used for lameness screening [23]. While movement asymmetries are often associated with lameness, not all movement asymmetries can be attributed to pain. When movement asymmetries were evaluated in a similar population of horses with no reported lameness issues by the owner [12], 70% of horses had movement asymmetries above the thresholds used for the inclusion in this study. Similarly, majority of polo ponies [32], Quarter Horses [26] and racehorses in training [33] presented with movement asymmetries. However, as the horses in this study did not go through a full lameness workup, the cause of the asymmetries was unknown. Consequently, it is not possible to rule out that some horses might have had underlying orthopaedic conditions which could have influenced the results.

Ideally, the data collection should be carried out at the same location to eliminate any variation due to different properties of surface which are common between different locations [34]. However, this is not always practical when recruiting participants for equine studies. Consequently, the exact specification of 'soft' and 'hard' surface varied between locations but the classification as 'soft' or 'hard' surface was representative of what would be used during orthopaedic evaluation in the field; e.g. 'soft' surface was any arena surface regardless of properties. Nevertheless, the surface was kept consistent for each horse, i.e. each horse was evaluated on the same soft or hard surface across the conditions (for both straight line and lungeing).

## Conclusion

In the riding horses in full work included in this study, changes in stride duration influenced head and pelvis movement asymmetries, highlighting the importance of keeping trotting speed consistent across trials. In addition, pelvic asymmetries were also systematically influence by circular motion but the pattern was different for asymmetries related to hindlimb weightbearing deficits (PDmin) and for asymmetries related to hindlimb propulsion deficits (PDmax). When compared to the straight-line trot condition, the PDmin asymmetries increased and PDmax asymmetries decreased when the hindlimb to which the asymmetry was assigned to was on the inside of a circle. When the affected hindlimb was on the outside, PDmin asymmetries decreased but PDmax asymmetries did not change. Interestingly, the variation in a surface type or the circle size did not influence movement asymmetries. Consequently, keeping surface type and circle size consistent between multiple lungeing evaluations might not be as important as controlling trotting speed. Further studies of horses with confirmed clinical diagnoses are needed to provide evidence of how movement asymmetries for specific orthopaedic conditions are influenced by circular motion, the type of surface and speed.

## Supporting information

**S1 Table. Full dataset with movement asymmetry values of horses included in this study.** (XLSX)

## Author Contributions

**Conceptualization:** Eva Marunova, Elin Hernlund, Emma Persson-Sjödin.

**Data curation:** Eva Marunova, Emma Persson-Sjödin.

**Formal analysis:** Eva Marunova.

**Investigation:** Eva Marunova, Elin Hernlund, Emma Persson-Sjödin.

**Methodology:** Eva Marunova, Elin Hernlund, Emma Persson-Sjödin.

**Resources:** Emma Persson-Sjödin.

**Supervision:** Elin Hernlund.

**Visualization:** Eva Marunova.

**Writing – original draft:** Eva Marunova, Emma Persson-Sjödin.

**Writing – review & editing:** Eva Marunova, Elin Hernlund, Emma Persson-Sjödin.

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
