## [Decision Letter · Decision Letter 0]

23 Nov 2023

PONE-D-23-26830Effect of circle, surface type and stride duration on vertical head and pelvis movement in ridden horses with pre-existing movement asymmetries identified during a straight-line trotPLOS ONE

Dear Dr. Marunova,

Thank you for submitting your manuscript to PLOS ONE. After careful consideration, we feel that it has merit but does not fully meet PLOS ONE’s publication criteria as it currently stands. Therefore, we invite you to submit a revised version of the manuscript that addresses the points raised during the review process.

We look forward to receiving your revised manuscript.

Kind regards,

Chris Rogers

Academic Editor

PLOS ONE

Journal Requirements:

"The data used in this study were collected as a part of a study assessing the effect of oral administration of non-steroidal anti-inflammatory drugs (NSAID) on upper body movement asymmetries [18] and which was funded by The Swedish-Norwegian Foundation for Equine Research (grant number H1347029) and the Swedish Research Council Formas (grant number 2014-1200328225-26)."

"Data used in this study was originally collected for a previous study, Effect of meloxicam treatment on movement asymmetry in riding horses in training. doi:10.1371/journal.pone.0221117. Co-authors Elin Hernlund and Emma Persson-Sjodin co-authored this previous study. However, they were not the recipients of the grant that supported data collection for the previous study. Grants from The Swedish-Norwegian Foundation for Equine Research (grant number H1347029) and the Swedish Research Council Formas (grant number 2014-1200328225-26) supported collection of data for the previous study."

Additional Editor Comments:

Thank you for the submission. Please find attached the reviewers comments. Both reviewers have suggested major review. Please have a look at these, complete the changes requested and I look forward to receiving the revised manuscript

Kind Regards

Chris

Reviewers' comments:

Reviewer's Responses to Questions

**Comments to the Author**

1. Is the manuscript technically sound, and do the data support the conclusions?

Reviewer #1: Partly

Reviewer #2: Yes

2. Has the statistical analysis been performed appropriately and rigorously? 

Reviewer #1: Yes

Reviewer #2: Yes

3. Have the authors made all data underlying the findings in their manuscript fully available?

Reviewer #1: Yes

Reviewer #2: Yes

4. Is the manuscript presented in an intelligible fashion and written in standard English?

Reviewer #1: Yes

Reviewer #2: Yes

5. Review Comments to the Author

Reviewer #1: This study builds on previous work to explore some of the practical challenges of measuring/analysing the results of vertical head and pelvic movement asymmetries during a clinical examination. Because the work is taken from another study I find the methods lacking and end up with a number of questions related to the research design that need clarification. The use of a soft surface would have been enhanced if the authors had defined ‘soft’ a little better. This distinction may also be a reason for the lack of significance of this factor. The discussion becomes rather long and unclear as it develops and I feel the clinical implications are largely not clinical implications, but further discussion. This section needs considerable revision. Currently the conclusion largely follows previous work. Perhaps the authors could consider how their analysis of data could be improved to pull out more meaningful outcomes.

Abstract

Too many results. Stick to the key results instead of including all of them and then explain what they mean instead of just writing a very general conclusion, which is really just another way of stating the results.

Line 70-71: Increased on a hard surface compared to what?

Line 71: the not this, and remove s

Line 71-73: Is this in a straight line? Please clarify.

Line 73: why potential?

Line 76 to 77: Horses may trot at different speeds or something like that instead of the speed might vary.

Line 79: Also, why is this? How was the difference in speed explained by these authors?

Line 90 to 92: Technically these are factors related to lunging compared to straight line motion asymmetries. No sure that ‘exercise related’ is really the right way to describe your factors.

Line 93: H1. What do you mean by direction and where is this justified as a hypothesis in your introduction?

Line 95: H2. You have already stated this has been found in Lines 51 to 54 from previous studies, so I am not sure how this hypothesis contributes additional knowledge?

Line 117-118: Reference for stride splitting?

Line 121 to 126: Please expand on your sign convention, so it is very clear when asymmetry is increased.

Line 129: Not clear in the introduction why circle diameter was introduced into the design.

Line 131: Define what you mean by fast

Line 136: More information is needed about the ‘soft’ surfaces. How soft was soft?

Line 137 to 140: If there were 16 conditions and the horses were already identified as having asymmetries in a straight line, how did you measure the conditions? How do you know that the increases in asymmetry are not a result of the order in which you collected the data?

Line 142: The inclusion criteria is somewhat confusing. You have said that the horses were in full training and considered sound by their owner. Then you go on to state that any judged as 2 or more out of 5 were excluded and that all had movement asymmetries above the Lameness Locator thresholds. Were the horses still in training on the day they came in for the testing and the owners considered them to be sound when they came in? Or did they come in for a veterinary assessment because they had become asymmetric/lame and were then included in the study if the owner agreed? This is very important. Please revise to clarify.

Line 150: Referring to another study related to the trial days and how each horse was grouped is not appropriate here, as it raises many more questions. Did the movement asymmetry change between days for straight line trot? If so, this suggests that these horses were not consistently asymmetric, which is quite different to a horse being consistently asymmetric (which would suggest sub-clinical or clinical pathologies). Non consistent asymmetry might not provide you with relevant results.

How many days were between trial days?

If horses had more than one predominant asymmetry, were they unilateral or multi-limb lame, as this may affect your results?

Please be very clear about what you did.

Line 175 to 181: Why? I don’t follow your reasoning here (as above).

Line 202: Not sure direction is the best description for inside-outside limbs on a circle/circle size/straight line. I am also wondering why straight is here. I thought you wanted to compare the difference between other conditions and straight?

Line 334-335: Your description of ‘movement direction’ which relates to inside/outside limbs probably needs to state inside/outside limb (where relevant) throughout the discussion for clarity.

Line 398 to 402: This statement (and actually Figure 2) might suggest that in order to unpick changes due to exercise condition related to head movement you may have been better to evaluate the difference between straight line and the other conditions.

Line 406: Inside and outside hindlimbs?

Line 454 – 456: Again, it depends on what the meaning of a ‘soft’ surface is.

Line 476 to 482: It is not clear what you are trying to say here. Please revise.

Line 483-503: It is very unclear what points you are trying to make here. This section is too long and lacks focus.

Line 513 to 529: What has this got to do with clinical implications? The only thing I am taking home from that sat the moment is that you are suggesting that horses with greater movement asymmetries/lameness should be trotted faster when they are assessed. Is this really your take home message? In addition, please be careful when reporting on studies that have found differences (in speed for example) with sound horses at much faster speeds compared to lame horses.

Line 530 to 547: If head asymmetry can change by up to the threshold, then surely that could mean that a horse trotted faster could exceed the threshold compared to the same horse that was trotted slower where a 3 mm difference was found? Would it not be a good idea to look at improving how speed is regulated?

Conclusion is rather long and mainly supports other work without really pulling out new and important information. Consider revising.

Reviewer #2: REVIEW PLOS ONE CIRCLE

14 could this sentence be made less vague? Aka …different conditions, like circles and various surfaces, yet better understanding of how these alter movement asymmetries is needed.

17 how did you control changing stride duration? Looking for this later, also were horses asymmetric for the same or different reasons? The underlying cause of the asymmetry is potentially influential

18 What discipline?

19 determined via subjective or objective measures? A grade 1 is very inconsistent so how can we know it is consistent enough to be present reliably under the new conditions?

24 what was the range? 6 mm to what? 3 mm to what? what did you classify as a grade 1?

25 to right front or hind limb? Or ?

69-71 confusing repetitive sentence and not sure what the s is at the end? Size?

121 why didn’t you use the Q score for the head?

129 how did you ensure the diameter ?

131 what were the cut offs to distinguish between slow and fast? A 10 percent difference? More?

136 how many on dirst vs aphalt?

144 is this the aaep scale? Reference?

146 unclear what at the discretion of the study vets meant? Were they or weren’t they allowed?

147 how far out from an NSAID were they?

164-170 this font seems smaller?

164 I thought they weren’t looked at while actively getting the NSAID? 167 I think you can leave the NSAID time point out and just discuss 3 times, else it is confusing

167-170 is not a complete sentence

104 need a section on where the horses came from? State the N even if in the other paper need some basic details here. I’m expecting breed, discipline, and more on their lameness in the results

189 unclear what inverted means? You used it twice vs explaining it , are we talking side? Or positive or negative? Or a fraction?

Results

Need descriptors for the animals, some indication of discipline, site of lameness may cause them to behave in different ways, signalment, breed etc.

Did you assess for normality?

Table 1 put the data regarding slow st line etc above the table

PD min seems to go below 3 when you look at the SD?

234 was stride duration really the same front and hind ?

Did you see statistically sig dif in stride duration bweteen conditions? P balues?

244 not clear what direction means? Inside or outside as you removed right and left?

Please add N to your tables

244 to 252 can you add in some comment about how it affected things? Like what you did in lines 259-262?

278-282 this sentence is unclear. What is smallest influence mean? Was it significant still ?

267 assuming the journal wants a smaller font for the fig?

278 see above re smallest influence etc.

296 I think the in interaction needs to be placed elsewuere to make more sense? Influenced by the interaction of direction and stide duration.

Please confirm all trotted and there were no pacers.

560 I think the term movement direction is unclear and misleading if it is truly straight line vs circling. Can you find a better indicator for it?

Did you run a power analysis to see how many animals you needed before hand?

For the discussion there are no limitations described. please add

Some commentary should be addressed to the cause of the asymmetry and what impact that can have on the parameters. WHat if there is bilateral hind limb lameness - what might that look like while circling etc.

6. PLOS authors have the option to publish the peer review history of their article (what does this mean?). If published, this will include your full peer review and any attached files.

Reviewer #1: No

Reviewer #2: No

---

## [Author Response · Author response to Decision Letter 0]

15 Mar 2024

All responses are provided in the 'Response to Reviewers' document.

---

## [Decision Letter · Decision Letter 1]

22 Apr 2024

PONE-D-23-26830R1Effect of circle, surface type and stride duration on vertical head and pelvis movement in ridden horses with pre-existing movement asymmetries identified during a straight-line trotPLOS ONE

Dear Dr. Marunova,

Thank you for submitting your manuscript to PLOS ONE. After careful consideration, we feel that it has merit but does not fully meet PLOS ONE’s publication criteria as it currently stands. Therefore, we invite you to submit a revised version of the manuscript that addresses the points raised during the review process. Thank you for the edits to the manuscript. The changes have improved the manuscript, but the reviewer have provided some additional changes / suggestions for the manuscript. Please have a look at these, as a number of these would improve the flow and readability of the manuscript.  Please submit your revised manuscript by Jun 06 2024 11:59PM. If you will need more time than this to complete your revisions, please reply to this message or contact the journal office at plosone@plos.org. Please include the following items when submitting your revised manuscript:A rebuttal letter that responds to each point raised by the academic editor and reviewer(s). You should upload this letter as a separate file labeled 'Response to Reviewers'.A marked-up copy of your manuscript that highlights changes made to the original version. You should upload this as a separate file labeled 'Revised Manuscript with Track Changes'.An unmarked version of your revised paper without tracked changes. You should upload this as a separate file labeled 'Manuscript'.

We look forward to receiving your revised manuscript.

Kind regards,

Chris Rogers

Academic Editor

PLOS ONE

Reviewers' comments:

Reviewer's Responses to Questions

**Comments to the Author**

1. If the authors have adequately addressed your comments raised in a previous round of review and you feel that this manuscript is now acceptable for publication, you may indicate that here to bypass the “Comments to the Author” section, enter your conflict of interest statement in the “Confidential to Editor” section, and submit your "Accept" recommendation.

Reviewer #2: (No Response)

Reviewer #3: (No Response)

2. Is the manuscript technically sound, and do the data support the conclusions?

Reviewer #2: Partly

Reviewer #3: Partly

3. Has the statistical analysis been performed appropriately and rigorously? 

Reviewer #2: Yes

Reviewer #3: Yes

4. Have the authors made all data underlying the findings in their manuscript fully available?

Reviewer #2: Yes

Reviewer #3: Yes

5. Is the manuscript presented in an intelligible fashion and written in standard English?

Reviewer #2: No

Reviewer #3: Yes

6. Review Comments to the Author

Reviewer #2: Effect of circle, surface type and stride duration on vertical head and pelvis movement

3 I’d leave off the identified during st line at the trot in the title- this is confusing as you say effect of circle…. Then straight line trot. I’ve fine with just pre-existing movement asymmetries.

14 unclear what upper body movement asymmetries is? This makes it sound like you mean a lameness of the shoulder, hip or pelvis. I’d remove upper body minimally and work to clarify this

18 I’d make mention of pre-existing asymmetries here somehow. After pelvis would seem appropriate

20 unclear what predominant asymmetries is? Do you mean front vs hind limb or objective degree of asymmetry? Aka the more lame together and the less lame/asymmetric together?

23 surfaces

24 for later, how was slow and fast set? How were these compared between horses in real time?

40 do we have diagnoses for these horses? Aka were all OA or all soft tissue? Before making a claim that surface doesn’t matter we would need to know this, as this statement can be misleading without that knowledge

48 looks like an extra space

65 a hard surface

66 a soft surface

71-72 how is your study different then if this has been established?

77 any pd min results?

I’d clarify if these asymmetries are considered clinically relevant by veterinarians even if the clients didn’t think so.

123 did you look at the q score?

162 did predominant asymmetries change between days? If so more details are needed. How many changed from front to hind etc if not reported below

Esp did asymmetries change if on the circle from the main one seen at the straight line?

214- what was the magnitude of the difference between the two? How did that range between horses? Was any horse fast as a default? How did they slow down with time?

219 how did you maintain either a 10 or 15 m circle assuredly?

221 how different were the surface types and how many different types were there (aka how many barns were evals performed at? Were evals at the same barn performed in the same spot)? Surface differences should be mentioned in limitations.

245 do you have BCS scores? Breeds? Lameness eval diagnoses?

268 I’d re mention what stride duration is a proxy for here

362-363 without knowledge of the cause of the assymetries I don’t feel we can make this conclusion as broad sweeping as it is.

366 a summary sentence with speed should be made here aka the faster they go the more asymmetry or whatever the finding is.

383 here again, the LLis calling them lame, but are they clinically lame? This needs to be clarified. The owners thinking they are not lame is not sufficient.

402 more comparison to previous papers should be undertaken to compare outcome variables to your study

435-440 please address if these horses are clinically lame as determined by a veterinarian and if so if diagnoses are known. If you aren’t considering them clinically lame how was this determined? How can you say they aren’t clinically lame when objective data is beyond the norm? did they have a full lameness work up?

500 why do you think your results differed?

540 given all of this, since horses did not have a full lameness eval, statements regarding circle size etc should be always qualified. Why was a full lameness not performed? Were subjective scores assigned at all? Was there a clinical corroboration with the LL? I have seen the LL fail especially with how it uses the law of sides (aka saying it’s a hind limb lameness when the forelimb issue is clearly a real cause of pain when issues present on the same side).

547 how do you recommend this is kept the same? Use stride duration on objective LL? How far beyond the same can it be and still be ok, please discuss this further in the discussion and add in more in results as needed.

548 decreased

553 surface type needs qualification aka unknown primary cause of asymmetry

Reviewer #3: This study aimed to determine the effect of several conditions, relating to surface type, trot stride duration, lunging diameter, and movement type (straight line vs. circular motion), on upper-body movement asymmetry parameters. The study is interesting and relevant, particularly as it has the potential to inform clinical practice. It is obvious that the authors have made significant changes to this manuscript, which should be applauded. However, the manuscript is still far too long and repetitive, making it very difficult to follow and thus to draw out the most important/relevant information. Further, clarity and conciseness are an essential element for any study, but particularly for a study design like this where over 15 different conditions are presented. Unfortunately, this is often not the case with this study, which increases the difficulty in reading, following, and drawing conclusions from such a long and repetitive piece of writing. There are several grammatical errors (particularly comma use) and run-on sentences throughout, which must be addressed. The methods section left me with several unanswered questions and there are several methodological limitations of this study which I do not feel have been adequately addressed/highlighted to the reader. I have attempted to highlight my more specific concerns in the points below.

INTRODUCTION:

Line 50: do you mean to say that these movement asymmetries are the result of natural/physiological adaptations that occur when a horse moves on a circular path? If so, what might some of these asymmetries on the lunge be caused by (i.e. body angle lean?)? You get into this later, in line 60, but the paragraph could be rearranged for clarity.

Line 55: “inside” – please check grammar and spelling throughout.

Line 57: not consistent in what sense? Please concisely add more context for the reader here. Not consistent within or between studies?

Line 63: “so it is important”

Line 65: secondly to what? Evaluated for what? Hard surface is also often used (albeit less frequently) for lunging, please edit this sentence for clarity.

Line 68: “uneven weightbearing” is this the most accurate description for the source of HDMin asymmetry?

Line 71: “and a small increase” were these increases significant? Same for the next lines?

Line 76: what is meant by “greater changes”? Be specific so that the reader obtains the appropriate level of context/background knowledge from the introduction section.

Line 78: new paragraph.

Line 84: “multiple runs” – define what is meant by this. Some readers may not be familiar with some of the terms used in this field.

Line 86: run on sentence – please revise for clarity and check grammar.

Line 91: “influence”. Also another run on sentence.

Line 93: lunging and circular motion are used interchangeably, which is fine, but probably confusing for the non-familiar reader. It would be useful to describe lunging as an exercise where the horse moves on a circular path around a handler at the beginning of the introduction. Why is this an important part of the lameness work up? This is also relevant information to (concisely) include. In line 93, do you mean that circle diameter will have a greater influence on movement asymmetry than surface …. Or do you mean lunging vs. straight line motion? Clarity needed for the hypotheses.

Line 97: consider using increase instead of amplified.

M&M:

This section lacks flow and is very difficult to follow (for example putting data processing before any description of the included horses and data collection procedures). One simple way to address this is by re-organising the sections (and their content) for clarity. I would suggest re-arranging this section so that the subsections are in the following order: Inclusion Criteria and Data Collection (including protocols for number of trials/runs etc., and a brief description of the proprietary software and asymmetry outputs that are used to define the inclusion criteria) � Data Processing � Data Analysis � Statistical Analysis. Note that the “Allocation to predominant movement asymmetry groups” and “Data normalisation” sections should go under Data Analysis and/or Processing sections.

Line 140: were screened.

Line 146: please provide a reference for the scale. This sentence should be revised to state that qualified veterinarians clinically evaluated each horse and those that were subjectively deemed to have a lameness grade >2/5 were excluded from the study. Assuming this is true and that veterinarians assessed each horse. Include the initials of the veterinarians if they are also authors on this manuscript.

Line 150 – thresholds.

Line 148 – 152 – run on sentence. Difficult to follow, particularly for readers unfamiliar with these thresholds/methods. Please revise for clarity.

I would argue that line 152 – 157 is not necessary, as it does not relate to the data used in this study. Line 157 can be moved to the beginning of this section (before Briefly, in line 138) and the rest removed. You can refer to Figure 1 in this section too.

Line 159: data were (data = plural).

Line 161 – 165: confusing and requires clarity. The irrelevant information about the larger study can be removed for clarity. In lines 159 – 163 it almost reads like all 4 time pointes were used in this study, as the 4-day washout period ensured that any influence of NSAID treatment on movement asymmetry was eliminated. Then you go on to say that actually only 3 of these timepoints were used for this study, so the above lines seem totally irrelevant/confusing (especially as we have Figure 1 and the reference for the larger study to refer to if needed). Remove Lines 159 – 163 and simply state that data from three trial days (PP, P, P-NSAID) were included for screening for predominant movement asymmetries, as this ensured that any treatment effect of NSAIDS were eliminated from this study. What do you mean by “up to three trial days” in line 163: does this mean that not all horses underwent screening over 3 days?

Did horses undergo screening for “predominant movement asymmetry” on the same day as data collection for this study? This needs clarification, as the actual procedures for data collection for the current study are very unclear. If data were not collected on the same day then how long between screening and data collection? The authors mention that nearly half of the horses included in this study did not consistently exhibit the same movement asymmetry across screening days, so how might this have influenced data grouping/analysis/interpretation if data were not collected on the same day (i.e. how could you be sure that this horse wouldn’t exhibit a different predominant movement asymmetry on the actual day of data collection)? More context needed here and this does need to be discussed as a limitation of your study if data were not collected on the same day as screening, particularly for horse without a consistent predominant asymmetry.

Time points and trial day are used interchangeably – be consistent. Are these referring to the same thing? What is meant by trial mean? How many straight line “runs/trials” were conducted per horse? Was this standardised and was the distance/surface/handler standardised across horses for screening?

Line 166: “greatest asymmetry” could the authors be certain that this was in fact related to a primary asymmetry or could this have been compensatory movement that resulted in greater asymmetry (i.e. primary hindlimb asymmetry exhibiting a greater compensatory asymmetry in head vertical displacement). If not, how may this have influenced some of your findings once the horse was evaluated across conditions? Is a compensatory asymmetry more likely to change across conditions than the primary source of asymmetry? If so, are your results valid? I realise that this is a difficult question to answer and that the answer is likely reliant on a full lameness evaluation or using induced lameness models, but this is definitely something that should be discussed in your limitations/discussion section.

Line 174: “data were”

Line 179: HDMin “mean difference across run/trail between the two halves of a stride for vertical displacement…..”? revise this definition, it us cumbersome and confusing.

Line 187: So did you exclude useful data from horses that showed consistent predominant movement asymmetries? Surely these data are key to truly understanding the effect of the different conditions studied here on horses that present with a consistent movement asymmetries, but yet you underrepresent these horses? These additional data seem invaluable and suggest that your dataset is saturated by data from horses with inconsistent movement patters/asymmetry, which increases the complexity of understanding the effects of these different conditions. Please explain. I may be confused, as I’m still unsure as to whether these IMU data were actually included in your analysis or were just used for grouping horses. This requires clarification.

Line 210: this information needs to come sooner in the methods section (see my suggestions above). Again – are these data the same as the “screening data”? Were they collected at the same time/day? If not, this information needs to be included in the article.

213: markers placed on the lunge to denote where the handler should hold the line? How were the handlers instructed to trot the horses faster? What was the subjective indicator of this so that it was kept consistent across horses?

Line 232 – these conditions should be defined and this should be done earlier in the methods section. For example: the direction condition is inconsistently defined. In line 212 and 227 you refer to left and right rein, but here it is inside/outside. Please clarify/be consistent throughout.

RESULTS

Line 250 – “one or more”. As stated above, nearly half of the horses showed inconsistent movement asymmetries. It would be interesting to know the proportion within of each Group of horses with consistent asymmetries and horses that were also included in one or more other groups. Were there patterns here? i.e., did some groups/asymmetry parameters have more horses with consistent asymmetry and did other groups have a greater proportion of horses included in 1 or more other groups? How might this have influenced your findings?

Line 255: the text should not repeat information provided in tables or figures, so line 255 – 257 can be removed. Lines 264 – 265 also repeat information provided in the text – is this relevant information for a table footer? Please review this throughout as the results section is currently too long.

Mean ± SD stride duration data should be placed in a table and referred to in the text. By putting these values in a text, the authors can remove lines 267 – 274 and keep only lines 274 – 278 (referring to the table). This will help to shorten this already too lengthy section.

Throughout – there should be a space between values and units (i.e. 714 ms). Consider presenting stride duration in s, which is arguably the norm. Descriptive stats should be presented as 714 ± 46 ms (example from line 268).

Line 278: mean range? What do you mean by this. Do you mean mean stride duration of 128 ms (range: 68 – 227 ms). Please check formatting for correctness throughout. Did you run any statistics on stride duration data between conditions? Surely this is important information. Was stride duration affected by circle diameter? It seems like there is some picking and choosing of what conditions are presented and what conditions are not in results. This should be addressed.

Line 284: “if stride length stays unchanged” – as far as I can tell, this was not quantified, so can you confirm that this is the case? Do you really expect that the horses trotted at a steady state and did not alter their stride length across these conditions? If not, how can you confidently infer that these horses actually trotted at a faster speed when interpreting your findings and clinical implications? I would be very cautious in making these statements, or at the very least, let the reader know that they should be cautious when interpreting your findings related to “trot speed”.

Line 286 and 323: what is meant by “on average horses altered their stride duration by 128 ms between the different exercise conditions”? this the average difference in stride duration between conditions? This is where a table for stride duration descriptive statistics would be helpful. What conditions can I expect to see this average increase in stride duration, and the resultant increase(?) in HDmin and PDMin? More information required – again, the table for stride duration data would be very useful.

Lines 299 – 303: run on.

Line 305: why was this stride duration chosen?

Line 306: “limb to which the predominant asymmetry was attributed to on the inside” this is repetitively used throughout the manuscript and is such a lengthy description. Consider including this definition in the methods section and then using a shorter description like “affected/assigned (?) limb on inside or outside of circle”. It would make for better readability throughout.

Line 317: re-word “significant interaction revealed that PDMin was affected by..”

Doesn’t line 321 repeat line 318?

DISCUSSION:

The discussion section is far too long, which makes it very difficult to identify the main take-away messages from this work. Please make this section more concise and try to focus on significant findings instead of discussing every variable/condition.

Line 355 – 358: run-on. Please revise so that the introductory sentence is a clear, concise description of what you did.

Lines 358 – 364 do an excellent job of summarising the main findings. However, because you did not directly measure trot speed, I would suggest altering lines 364 – 366 to state that “changes in stride duration resulted in significant differences in X, Y, Z movement asymmetries, which suggests that handlers should endeavour to keep trot speed consistent during clinical evaluations”.

Line 376: why might this be? Please use the discussion section to identify reasons for why your findings agree with, or differ from, other studies. Were there methodological reasons that might explain these differences between studies? Line 383, are you inferring that the movement asymmetries in your study are due to laterality and normal biological variation and this is why you didn’t see significant differences during lunging conditions? Can you be sure that this is the only reason, as horses were not clinically assessed for lameness? More detail needed and methodological limitations of this study should be considered when interpreting findings/comparing with other studies.

Line 380: “increases in head movement asymmetry were not identified”.

Line 388: “higher peace vertical forces increased load placed on affected structure”? please revise.

Line 394: “suggests”. How might they be affected differently by changes in stride duration/”speed”?

Line 396: please be sure to clearly state where results are drawn from previous studies, as it is sometimes hard to differentiate between discussion of your own findings and those of others. “Previous studies have demonstrated and effect of surface type on trot speed….”

Line 401: as “stride duration and speed” – please be clear that you only measured stride duration and are making inferences about speed from these data.

Line 407: repetitive, please remove.

Line 413: why is this?

Line 418/419: “decreasing vertical displacement of the pelvis” instead of “lower rise”. This line needs a reference. Line 422: asymmetric hindlimb propulsion.

Lines 424 – 427: run on sentence and repeats results. Revise.

Lines 434 – 446: Caution should be taken when stating that surface type has no influence on pelvic movement asymmetries, especially given the limitations of this study where a variety of surfaces were used/not standardised, which will undoubtedly have influenced your findings. Please make this clear to the reader and avoid sweeping statements. Lines 436 – 440 – run on sentence, revise. Doesn’t line 441 contradict your statements in the previous paragraph?

Line 449: “significant”.

Line 452: “moderate”.

Line 456: “influences”

Line 455: The authors need to consider the influence of steady state treadmill vs. overground locomotion when making this type of claim. Please revise as it is a sweeping statement.

Line 466: So what can we draw from these findings in relation to stride duration and pelvic movement asymmetries? The take home message is unclear.

“Implications for Clinical Practice” – consider removing this section. It is highly repetitive (of results and previous parts of the discussion section) and will help to make your discussion section more concise. Further, and importantly, is it wise to extrapolate the results of this study into clinical practice? There are several methodological limitations that influence the validity of these findings and surely more standardised work is required before clinical recommendations can be made.

Line 472 “in clinical practice”.

Line 476: very large variation, hence my point above.

Lines 474 – 487: repeats what has already been presented in results and discussion. The question of “how tightly does speed need to be controlled in a clinical setting” is never answered. Can you make this conclusion based on your measurements in this study?

Line 490: Second to what? Again, this paragraph is repetitive and does not provide recommendations for clinical practice. Please remove.

Line 510 – 518 is well written and could be incorporated elsewhere in the discussion section.

Limitations: several of the main limitations of this study are not addressed in this section.

Line 524: these are mainly treadmill studies where velocity is standardised. The limitations related to extrapolating changes in stride duration with speed during overground locomotion need to be clearly highlighted to the reader, especially if you’re going to recommend that SD could be a proxy for speed (which I would recommend removing).

The lack of standardisation of surfaces, stride velocity etc need to be highlighted as limitations of this study as these are confounding factors that will undoubtedly have influenced your results. Further, my above concerns regarding the groupings/analysis of horses with inconsistent/vs consistent movement asymmetries should be addressed in this section.

Conclusion:

Line 548: “decreased”. Lines 547 – 552: extremely vague – does not leave the reader with a good take-away message relating to how movement asymmetries are affected by “circle direction”. Please look to the conclusions drawn in the abstract and try to incorporate these in the conclusions section to improve the take-away message.

7. PLOS authors have the option to publish the peer review history of their article (what does this mean?). If published, this will include your full peer review and any attached files.

Reviewer #2: No

Reviewer #3: No

---

## [Author Response · Author response to Decision Letter 1]

30 Jul 2024

All responses are included in the attached document.

---

## [Editor Report · Decision Letter 2]

5 Aug 2024

Effect of circle, surface type and stride duration on vertical head and pelvis movement in riding horses with pre-existing movement asymmetries in trot

PONE-D-23-26830R2

Dear Dr. Marunova,

We’re pleased to inform you that your manuscript has been judged scientifically suitable for publication and will be formally accepted for publication once it meets all outstanding technical requirements.

Kind regards,

Chris Rogers

Academic Editor

PLOS ONE

Additional Editor Comments (optional):

Thank you for the edits to the manuscript. The changes have greatly increased the clarity and the quality of the manuscript. I am happy for the manuscript to now proceed with the publication process.
---

## [Editor Report · Acceptance letter]

8 Aug 2024

PONE-D-23-26830R2 

PLOS ONE

Dear Dr. Marunova, 

I'm pleased to inform you that your manuscript has been deemed suitable for publication in PLOS ONE. Congratulations! Your manuscript is now being handed over to our production team.

Kind regards, 

on behalf of

Dr. Chris Rogers 

Academic Editor

PLOS ONE